# INFLLM-V2: DENSE-SPARSE SWITCHABLE ATTENTION FOR SEAMLESS SHORT-TO-LONG ADAPTATION

**Weilin Zhao[1], Zihan Zhou[2], Zhou Su[2], Chaojun Xiao[1]\*, Yuxuan Li[2], Yanghao Li[1],**
**Yudi Zhang[3], Weilun Zhao[2], Zhen Li[2], Yuxiang Huang[1], Ao Sun[2], Xu Han[1]\*, Zhiyuan Liu[1]\***
[1]Tsinghua University  [2]OpenBMB  [3]Harbin Institute of Technology
zwl23@mails.tsinghua.edu.cn  {xcj,han-xu,liuzy}@tsinghua.edu.cn

## ABSTRACT

Long-sequence processing is a critical capability for modern large language models. However, the self-attention mechanism in the standard Transformer architecture faces severe computational and memory bottlenecks when processing long sequences. While trainable sparse attention methods offer a promising solution, existing approaches such as NSA introduce excessive extra parameters and disrupt the conventional *pretrain-on-short, finetune-on-long* workflow, resulting in slow convergence and difficulty in acceleration. To overcome these limitations, we introduce dense-sparse switchable attention framework, termed as InfLLM-V2. InfLLM-V2 is a trainable sparse attention that seamlessly adapts models from short to long sequences. Specifically, InfLLM-V2 reuses dense attention parameters through parameter-free architecture modification, maintaining consistency between short and long sequence processing. Additionally, InfLLM-V2 ensures computational efficiency across all sequence lengths, by using dense attention for short inputs and smoothly transitioning to sparse attention for long sequences. To achieve practical acceleration, we further introduce an efficient implementation of InfLLM-V2 that significantly reduces the computational overhead. Our experiments on long-context understanding and chain-of-thought reasoning demonstrate that InfLLM-V2 is $4\times$ faster than dense attention while retaining 98.1% and 99.7% of the performance, respectively. Based on the InfLLM-V2 framework, we have trained and open-sourced MiniCPM4.1[1], a hybrid reasoning model, providing a reproducible implementation for the research community.

## 1 INTRODUCTION

With the rapid development of large language models (LLMs) (Brown et al., 2020; Bommasani et al., 2021; Han et al., 2021; OpenAI, 2023), the demand for long-sequence processing capabilities has become increasingly critical. From long-input scenarios such as deep research (Zheng et al., 2025; Xu & Peng, 2025), chatbots with long-term memory, and software issue resolution (Jimenez et al., 2023; Yang et al., 2025), to long-output tasks including complex reasoning (OpenAI et al., 2024; DeepSeek et al., 2025) and LLM-driven agents (Wang et al., 2024), a model's capability to understand and generate long sequences directly determines its performance in real-world applications. However, the self-attention mechanism in the existing Transformer (Vaswani et al., 2017) architecture faces severe computational and memory bottlenecks when processing long sequences.

To address the challenge of processing long sequences, efforts have been devoted to exploring sparse attention mechanisms (Beltagy et al., 2020; Zaheer et al., 2020; Tay et al., 2022), which restrict each token within the context to attend to only a subset of tokens related to that token. Early research in this area focuses on the training-free setting, leveraging the sparsity naturally occurring in self-attention mechanisms to accelerate inference (Xiao et al., 2024a;b; Jiang et al., 2024). However, the training-free setting introduces a fundamental trade-off between sparsity and model performance. To avoid significant performance degradation, the degree of sparsity that can be applied is often limited, which in turn restricts the potential efficiency gains.

---

\*Corresponding Authors.
[1]https://huggingface.co/openbmb/MiniCPM4.1-8B

Figure 1: The comparison of Vanilla Full Attention, NSA (Yuan et al., 2025), and our InfLLM-V2.

Given the limitations of training-free attention mechanisms, trainable sparse attention mechanisms have garnered increasing attention from researchers (Lu et al., 2025; Gao et al., 2024). Among them, the natively trainable sparse attention (NSA) (Yuan et al., 2025) method adopts the widely-used block-sparse attention (Child et al., 2019) structure, designing three different sparse attention modules and developing corresponding CUDA kernels to accelerate model computation. Despite its effectiveness, we find **misalignment between the sparse architecture of NSA and the standard pretrain-on-short, finetune-on-long workflow**. A widely used way to build long LLMs is to pretrain on short sequences and finetune on long sequences. The NSA creates an architectural mismatch with vanilla full attention, as it introduces three sets of key-value parameters and three attention modules, forcing the model to abruptly switch from a single-output attention to a multi-output attention architecture. As shown in Section 4, this mismatch hinders smooth adaptation, erases what the model has already learned, and introduces efficiency bottleneck for short sequences.

To address all the above issues, we propose dense-sparse switchable attention framework (**InfLLM-V2**). InfLLM-V2 is built on InfLLM (Xiao et al., 2024a), a training-free block-sparse attention mechanism, and introduces three core innovations:

1. **Seamless Short-to-Long Adaptation:** As depicted in Figure 1, different from NSA, which requires additional parameters and multiple attention modules, InfLLM-V2 seamlessly transitions from dense to sparse attention by directly reusing existing dense attention parameters. This design naturally aligns with the standard pretrain-on-short, finetune-on-long workflow, eliminating architectural mismatches and minimizing loss fluctuations.

2. **Efficiency for Both Short and Long Sequences:** Because the transition from dense to sparse attention in InfLLM-V2 requires no additional parameters and introduces minimal distributional shifts, the model preserves its strong performance on short texts and can easily switch back to dense attention for short sequence efficiency.

3. **Accelerated Block Selection Mechanism:** The block selection step before sparse attention inherently undermines the efficiency gains of the sparse attention itself. We propose a hardware-aware efficient implementation, effectively removing the bottleneck and unlocking the full potential of sparse attention.

We evaluate our method on long-context understanding and long chain-of-thought (CoT) generation benchmarks. Our InfLLM-V2 is $4\times$ faster than dense attention while maintaining 98.1% and 99.7% of the original performance on these tasks, respectively. We release all associated implementations in `https://github.com/OpenBMB/infllmv2_cuda_impl`.

## 2 RELATED WORK

As the demand for LLMs to understand and generate long sequences continues to grow, research on improving attention efficiency has garnered increasing attention (Tay et al., 2022; Sun et al., 2025; Zhang et al., 2025a). In this section, we discuss the sparse attention paradigm from two perspectives: training-free and trainable sparse attention approaches.

### 2.1 TRAINING-FREE SPARSE ATTENTION

Training-free sparse attention approaches aim to utilize the intrinsic sparsity of attention layers. These methods enable LLMs trained with dense attention to perform sparse attention between each token and a small subset of relevant contexts. Based on the selection strategy for relevant contexts, these algorithms can be categorized into predefined sparse patterns and dynamic sparse patterns.

**Predefined Sparse Patterns.** Sparse attention with a predefined pattern employs manually defined heuristic rules to determine which contextual tokens should be selected for attention compu-

tation (Xiao et al., 2024b; Han et al., 2024; Child et al., 2019; Zaheer et al., 2020; Beltagy et al., 2020; Xiao et al., 2025). For instance, sliding window attention restricts each token to interact only with neighboring tokens (Beltagy et al., 2020). Building upon sliding windows, some works select special tokens such as initial tokens or segment separators, requiring all tokens to attend to these special tokens (Xiao et al., 2024b; Chen et al., 2024; Child et al., 2019). Furthermore, some works combine multiple predefined attention patterns (Zaheer et al., 2020; Beltagy et al., 2020). These approaches typically rely on human observations to formulate heuristic rules for selecting relevant contexts.

**Dynamic Sparse Patterns.** Dynamic sparse patterns incorporate the semantic information of query tokens into the context selection process by computing the relevance between query tokens and candidate contexts. Early works primarily perform similarity computation at the token level (Kitaev et al., 2020; Roy et al., 2021; Wang et al., 2020). As sequence lengths increase, block sparse methods have gained widespread adoption, which partition contexts into contiguous block units and perform relevance computation and context selection at the block granularity (Xiao et al., 2024a; Jiang et al., 2024; Xu et al., 2025; Tang et al., 2024; Zhang et al., 2025b; Lai et al., 2025). Furthermore, research on attention sparsity has inspired the development of key-value (KV) eviction and compression methods, which reduce memory consumption by discarding or compressing KV caches with low attention probabilities (Zhang et al., 2023; Li et al., 2024; Huang et al., 2024; 2025).

Training-free methods, while focusing on improving the inference efficiency of dense attention models, are often constrained by insufficient sparsity levels in order to avoid severe performance degradation and finally suffer from limited acceleration benefits.

## 2.2 TRAINABLE SPARSE ATTENTION

To further enhance efficiency for long sequence processing, researchers incorporate sparse attention into the model training phase. SeerAttention (Gao et al., 2024) employs a self-distillation post-training algorithm to train a router that selects relevant contexts for query blocks. MoBA (Lu et al., 2025) employs a block sparse attention structure during the short-to-long adaptation phase, training routers between query blocks and KV blocks for context selection. These methods partition query tokens into blocks and can only accelerate the prefilling phase. NSA (Yuan et al., 2025) designs three attention components for token-level sparsity, effectively accelerating both prefilling and decoding processes. However, NSA introduces substantial additional parameters, making it unsuitable for efficient short-to-long adaptation and imposing significant computational overhead on short-sequence processing. In this paper, we focus on proposing a sparse attention mechanism that effectively and efficiently processes both short and long sequences, supporting both prefilling and decoding.

## 3 METHOD

### 3.1 BACKGROUND

**Grouped-Query Attention.** Attention mechanisms enable models to selectively focus on relevant parts of the input sequence. Among various attention variants, grouped-query attention (GQA) (Ainslie et al., 2023) has emerged as a popular method that strikes a balance between model performance and computational efficiency. Given an input sequence of hidden states $\mathbf{X} \in \mathbb{R}^{n \times d}$, where $n$ is the sequence length and $d$ is the model dimension, GQA computes the queries ($\mathbf{Q}$), keys ($\mathbf{K}$), and values ($\mathbf{V}$) via linear projections as $\mathbf{Q} = \mathbf{X}\mathbf{W}_Q, \mathbf{K} = \mathbf{X}\mathbf{W}_K, \mathbf{V} = \mathbf{X}\mathbf{W}_V$. The projection matrices have the shapes $\mathbf{W}_Q \in \mathbb{R}^{d \times (h_q d_h)}$ and $\mathbf{W}_K, \mathbf{W}_V \in \mathbb{R}^{d \times (h_{kv} d_h)}$, with the head dimension $d_h$. These tensors are then reshaped to form $h_q$ query heads $\{\mathbf{Q}_i\}_{i=1}^{h_q}$, $h_{kv}$ KV heads $\{\mathbf{K}_j, \mathbf{V}_j\}_{j=1}^{h_{kv}}$, with each head having the shape $n \times d_h$. The query heads are partitioned by a group size $G = h_q/h_{kv}$. The attention scores $\mathbf{S}_i$ and the attention output $\mathbf{O}_i$ for the $i$-th query head are computed by attending to its corresponding KV heads with the index $j = \lfloor (i-1)/G \rfloor + 1$:

$$\mathbf{S}_i = \text{Softmax}\left(\frac{\mathbf{Q}_i \mathbf{K}_j^\top}{\sqrt{d_h}}\right), \quad \mathbf{O}_i = \mathbf{S}_i \mathbf{V}_j. \tag{1}$$

The final output is obtained by concatenating the attention outputs and projecting them through a final linear layer $\mathbf{W}_O \in \mathbb{R}^{(h_q d_h) \times d}$: $\text{Attention}(\mathbf{X}) = \text{Concat}(\mathbf{O}_1, \ldots, \mathbf{O}_{h_q})\mathbf{W}_O$.

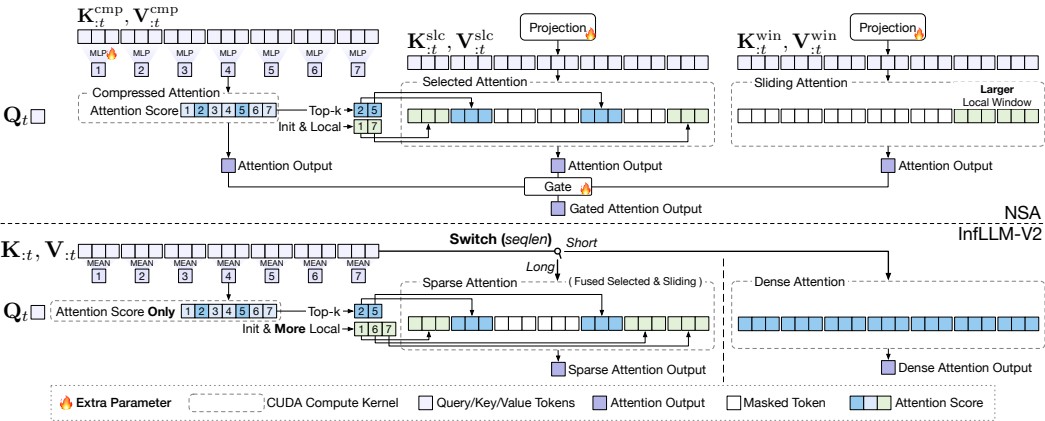

Figure 2: The overview of NSA and InfLLM-V2. InfLLM-V2 uses a shared KV for both Sparse Attention and Dense Attention. InfLLM-V2 fuses Selected Attention and Sliding Attention and eliminates the output of Compressed Attention. InfLLM-V2 introduces no extra parameters.

**NSA.** NSA (Yuan et al., 2025) is an enhancement of GQA designed for efficiency on long sequences. The key insight is that for long sequences, e.g., when $n > 32k$, the attention score matrix $\mathbf{S}$ exhibits strong sparsity. This allows for approximating the attention matrix by ignoring negligible values, leading to faster computation. As illustrated in Figure 2, NSA utilizes three distinct modules and combines them using a gating module. Based on the observation that adjacent attention scores are similar (Jiang et al., 2024), NSA splits the sequences into blocks of size $B$. First, *Compressed Attention* employs a compressed representation of the KV tensors to reduce the computational complexity. Second, *Selected Attention* leverages the attention scores from compressed attention to compute only the blocks with high attention scores. Finally, *Sliding Attention* is used to focus on local contextual information within the sequence. For these three attention modes, they introduce three sets of KV projection matrices: $\mathbf{W}_K^{\mathrm{cmp}}, \mathbf{W}_V^{\mathrm{cmp}}, \mathbf{W}_K^{\mathrm{slc}}, \mathbf{W}_V^{\mathrm{slc}}, \mathbf{W}_K^{\mathrm{win}}, \mathbf{W}_V^{\mathrm{win}}$. This final output can be mathematically represented as Output $= g^{\mathrm{cmp}}\mathbf{O}^{\mathrm{cmp}} + g^{\mathrm{slc}}\mathbf{O}^{\mathrm{slc}} + g^{\mathrm{win}}\mathbf{O}^{\mathrm{win}}$, where $\mathbf{O}^{\mathrm{cmp}}$, $\mathbf{O}^{\mathrm{slc}}$, and $\mathbf{O}^{\mathrm{win}}$ are the outputs of the three respective modules, and the gate scores $g^{\mathrm{cmp}}$, $g^{\mathrm{slc}}$, and $g^{\mathrm{win}}$ are derived from the input features $\mathbf{X}$ via an MLP and a sigmoid activation. They also train an MLP module for compressing the KV tensors. The three distinct KV projections, combined with an additional MLP and gating module, result in a highly complex architecture. This complexity, in turn, makes the model poorly suited for training from scratch on short-sequence data and also complicates the process of converting pretrained dense models to sparse ones.

## 3.2 OVERALL FRAMEWORK

We propose InfLLM-V2, a more concise framework with zero extra parameters that more closely aligns dense and sparse attention patterns.

**Shared Key-Value Projection.** We find that using three separate sets of KV projection parameters in NSA (Yuan et al., 2025) is unnecessary, which not only complicates the adaptation from short to long sequences but also significantly slows down computation for short sequences. Therefore, we propose using a single shared set of projection parameters, $\mathbf{W}_K$ and $\mathbf{W}_V$, initialized with the pretrained dense attention parameters and used for finetuning on long sequences.

**Aligned Computation.** In addition to ensuring that sparse and dense attention share the same parameters, their computational processes must also be closely aligned. In NSA, the three attention modules all generate outputs that are aggregated by an extra gating module. This forces the computation of all three modules even for short sequences, leading to substantial overhead. To mitigate this, we take a union of the two sparse patterns in *Selected Attention* and *Sliding Attention* and eliminate the output of *Compressed Attention*, forming a unified *Sparse Attention* module. Specifically, the original *Selected Attention* module identifies important token blocks based on the attention scores from the *Compressed Attention* module, $\mathbf{S}^{\mathrm{cmp}}$. For a query token with index $i$, located in the block $b_i = \lfloor \frac{i-1}{B} \rfloor + 1$, attention is always granted to a fixed set of initial blocks and a set of local blocks:

$$\mathcal{I}_{\mathrm{init}} = \{1, 2, \ldots, N_{\mathrm{init}}\}, \quad \mathcal{I}_{\mathrm{local}}(i) = \{b_i - N_{\mathrm{local}} + 1, \ldots, b_i - 1, b_i\}. \tag{2}$$

The top-k selection is then applied to $\mathbf{S}^{\mathrm{cmp}}$ over the set of remaining blocks, denoted as $\mathcal{I}_{\mathrm{topk}}(i)$. The complete set of attended block indices for this query token is the union of these three sets:

$$\mathcal{I}(i) = \mathcal{I}_{\mathrm{init}} \cup \mathcal{I}_{\mathrm{local}}(i) \cup \mathcal{I}_{\mathrm{topk}}(i). \tag{3}$$

If we denote the set of token indices in the $j$-th block as $T_j = \{jB+1, \ldots, (j+1)B\}$, the selected attention allows a token in the block $b_i$ to attend to the union of blocks $\bigcup_{j \in \mathcal{I}(i)} T_j$. The *Sliding Attention*, on the other hand, allows the $i$-th token to attend to a range $\{i-w+1, \ldots, i\}$ of window size $w$. Since the local blocks in *Selected Attention* and the window in *Sliding Attention* create overlapping, we merge them by expanding the number of local blocks within our unified *Sparse Attention* to strictly cover the region of the *Sliding Attention*, that is, $N_{local} \geq \lceil \frac{w}{B} \rceil + 1$, as illustrated in Figure 3.

Furthermore, we eliminate the output of the *Compressed Attention* module, only retaining its attention scores $\mathbf{S}^{\mathrm{cmp}}$ for block selection in *Sparse Attention*. This single-output design more closely mirrors dense attention and aids the training of the sparse attention model. InfLLM-V2 can thus dynamically switch between dense and sparse attention patterns based on the input sequence length.

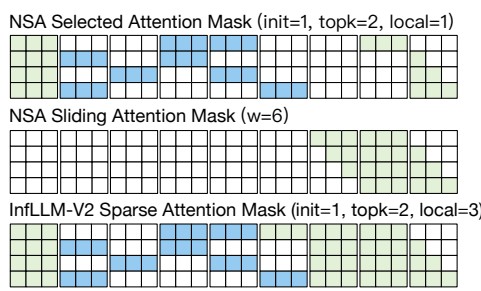

**Simplified and Efficient Compression Module.** Since we eliminate the output of the *Compression Attention*, using MLP for token

Figure 3: The illustration of the union of *Selected Attention* and *Sliding Attention*.

compression would not receive gradients. We replace it with a more intuitive parameter-free pooling function, which will be detailed in Section 3.3. Additionally, computing the attention scores $\mathbf{S}^{\mathrm{cmp}}$ introduces non-negligible overhead, and we will reduce this overhead in Section 3.4.

## 3.3 BLOCK REPRESENTATION

Simply compressing a long sequence with a large block size $B$ in 1-stage can lead to a significant loss of granular information (Yuan et al., 2025). To address this, we implement a 3-stage, coarse-grained to fine-grained compression process, as shown in Figure 4. In the first stage, we process the input key sequence $\mathbf{K}$ to produce an intermediate and coarse-grained representation $\mathbf{K}^{C_1}$. By denoting the initial compression block size as $l_{C_1}$ and the stride as $s_{C_1}$, we achieve this by applying a **mean-pooling** operation over sequential blocks:

$$\mathbf{K}_i^{C_1} = \mathrm{Mean}(\mathbf{K}_{i \cdot s_{C_1} : i \cdot s_{C_1} + l_{C_1}}). \tag{4}$$

Then, we compute the attention scores $\mathbf{S}^{C_1}$ between the query $\mathbf{Q}$ and $\mathbf{K}^{C_1}$:

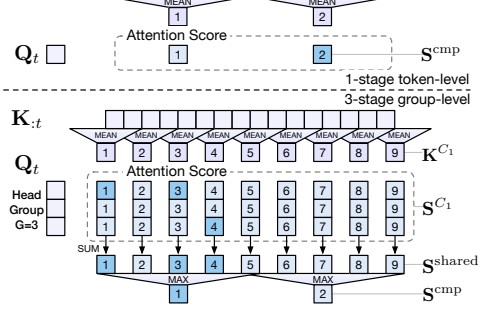

Figure 4: The illustration of the 3-stage group-level compression, compared with the 1-stage token-level compression.

$$\mathbf{S}^{C_1} = \mathrm{Softmax}(\mathbf{Q}(\mathbf{K}^{C_1})^{\top}). \tag{5}$$

In the second stage, we employ block-wise sparse attention rather than token-level approaches for the efficiency of *Sparse Attention*. In a model utilizing GQA, we can achieve this by forcing the block selection pattern across all heads within a group to be the same. We conduct **summation** within the head group to get the shared importance score $\mathbf{S}^{\mathrm{shared}}$:

$$\mathbf{S}^{\mathrm{shared}} = \sum_{h=1}^{G} \mathbf{S}^{C_1}(h). \tag{6}$$

In the third stage, we apply a **max-pooling** operation, which can preserve the most salient features. The aggregated score $\mathbf{S}^{\mathrm{cmp}}$ are defined as follows and used for the *Sparse Attention*:

$$\mathbf{S}_i^{\mathrm{cmp}} = \mathrm{Max}(\mathbf{S}_{i \cdot s : i \cdot s + l}^{\mathrm{shared}}). \tag{7}$$

---

**Algorithm 1** Computation of $\mathbf{S}^{\text{shared}}$ (Suppose $h_{kv} = 1$ without loss of generality.)

---

**Require:** $\mathbf{Q} \in \mathbb{R}^{n \times G \times d_h}$, $\mathbf{K}^{C_1} \in \mathbb{R}^{(n/s_{C_1}) \times d_h}$, $\mathbf{K}^{C_2} \in \mathbb{R}^{(n/s_{C_2}) \times d_h}$ in HBM. Block sizes $B_q, B_k$.

    Divide $\mathbf{Q}$ into $T_q = \lceil n/B_q \rceil$ blocks $\mathbf{Q}_1, \ldots, \mathbf{Q}_{T_q}$ of size $B_q \times G \times d_h$ each.

    Divide $\mathbf{K}^{C_1}$ into $T_1 = \lceil n/s_{C_1}/B_k \rceil$ blocks $\mathbf{K}_1^{C_1}, \ldots, \mathbf{K}_{T_1}^{C_1}$ of size $B_k \times d_h$ each.

    Divide $\mathbf{K}^{C_2}$ into $T_2 = \lceil n/s_{C_2}/B_k \rceil$ blocks $\mathbf{K}_1^{C_2}, \ldots, \mathbf{K}_{T_2}^{C_2}$ of size $B_k \times d_h$ each.

    Divide $\mathbf{S}^{\text{shared}}$ into $T_q \times T_1$ blocks of size $B_q \times B_k$ each.

    **for** $i = 1, \ldots, T_q$ (parallel) **do**

        Load $\mathbf{Q}_i$ from HBM to on-chip SRAM.

        On chip, initialize online-softmax related statistic log-sum-exp $lse$.

        **for** $j = 1, \ldots, T_2$ (sequential) **do**              ▷ First pass (Coarse-grained)

            Load $\mathbf{K}_j^{C_2}$ from HBM to on-chip SRAM.

            On chip, compute attention scores $\mathbf{S}_{ij}^{C_2} \in \mathbb{R}^{G \times B_q \times B_k}$ as in Eq. (8) and update $lse$.

        **for** $j = 1, \ldots, T_1$ (sequential) **do**             ▷ Second pass (Fine-grained)

            Load $\mathbf{K}_j^{C_1}$ from HBM to on-chip SRAM.

            On chip, compute attention scores $\mathbf{S}_{ij}^{C_1} \in \mathbb{R}^{G \times B_q \times B_k}$ as in Eq. (5) and normalize it using $lse$.

            On chip, compute the final block $\mathbf{S}_{ij}^{\text{shared}} \in \mathbb{R}^{B_q \times B_k}$ by summing $\mathbf{S}_{ij}^{C_1}$ over the head group.

            Write the block $\mathbf{S}_{ij}^{\text{shared}}$ to its corresponding position in HBM.

    **return** the output $\mathbf{S}^{\text{shared}}$.

---

In our method, we set $l_{C_1} = \frac{B}{2}$, $s_{C_1} = \frac{B}{4}$, $l = 5$, and $s = 4$ so that it can achieve the same compression ratio as 1-stage compression of block size $B$. Intuitively, we compute the sparse scores of the entire block based on several sliding sub-blocks within the block.

## 3.4 EFFICIENT IMPLEMENTATION

For efficient *Sparse Attention*, we follow the techniques in NSA (Yuan et al., 2025) to set the group size $G$ of GQA to 16, a configuration well-suited for block sparse attention. More details can be found in Appendix B. **However, our profiling reveals that the computation of the compression score, $\mathbf{S}^{\text{cmp}}$, introduces a significant performance bottleneck.** A primary source of this slowdown is the substantial I/O required to store the first-stage attention scores $\mathbf{S}^{C_1}$ into the slow GPU HBM. The amount of data that needs to be written is $h_q n^2 / s_{C_1}$, where $n$ is the full sequence length. Given that $s_{C_1} \ll n$, materializing the full attention score matrix to GPU HBM incurs a prohibitive cost.

Drawing inspiration from FlashAttention (Dao, 2024), we aim to minimize this I/O by ensuring the attention scores remain within the fast GPU SRAM as much as possible. Our approach, ***Fused Head Group Summation***, is to fuse the summation over the head group, required for the second-stage compression, directly into the SRAM-based computation loop of FlashAttention. After that, we can only store the reduced attention scores $\mathbf{S}^{\text{shared}}$ into GPU HBM, whose size is $h_q n^2 / (s_{C_1} G)$.

Another challenge arises from the fact that summing over the head group dimension and performing the online-softmax (Dao, 2024) along the sequence dimension are not commutative operations. This conflict prevents a straightforward fusion. To overcome this, we implement a two-pass approach. In the first pass, we compute the log-sum-exp ($lse$) term required for the softmax normalization within the SRAM. In the second pass, we leverage the $lse$ to calculate the final attention scores, perform the summation across the head group within the SRAM, and write the reduced scores to the HBM. The trade-off of this two-pass method is that it doubles the computational workload. Therefore, we propose ***LSE Approximation*** to approximate the $lse$ computation by using a coarser-grained attention score $\mathbf{S}^{C_2}$. Following Eq. (4) and Eq. (5), we change them to

$$\mathbf{K}_i^{C_2} = \text{Mean}(\mathbf{K}_{i \cdot s_{C_2} : i \cdot s_{C_2} + l_{C_2}}), \quad \mathbf{S}^{C_2} = \text{Softmax}(\mathbf{Q}(\mathbf{K}^{C_2})^\top). \tag{8}$$

By setting $s_{C_2} = 4s_{C_1}$ and $l_{C_2} = 4l_{C_1}$, the computational overhead was reduced from $2\times$ to $1.25\times$. We summarize the procedure for computing $\mathbf{S}^{\text{shared}}$ in Algorithm 1. To further reduce memory I/O, the max-pooling and top-k operations related to $\mathbf{S}^{\text{cmp}}$ could also be fused into the kernel; however, we leave this implementation for future work.

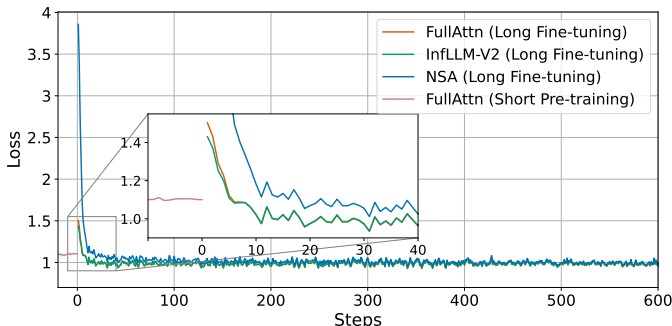

Figure 5: The training loss of models. We only show the last few iterations of the short pretraining.

# 4  EXPERIMENT

We evaluate InfLLM-V2 on tasks ranging from short to long contexts, and demonstrate its efficiency.

## 4.1  EXPERIMENT SETUP

**Pretraining Setup.**   We first use full attention to pretrain a model on short-sequence data, marked as SHORT. We employ a standard GQA (Ainslie et al., 2023) model backbone with 8B parameters, with the hidden size $d = 4096$, the number of heads $h_q = 32$, $h_{kv} = 2$, and the head dimension $d_h = 128$. The pretraining dataset consists of 8T tokens of 4k-length sequences, primarily comprising FineWeb-Edu (Penedo et al., 2024) and Stack-v2 (Lozhkov et al., 2024). We set 8M tokens per batch, and use a WSD learning rate scheduler (Hu et al., 2024) with 2000 warmup steps to an initial learning rate of 7.5e-3, and 27000 decay steps to the final learning rate of 3e-4.

**Long-Context Adaptation.**   When transitioning to long-context finetuning, we switch to INFLLM-V2 (SPARSE). Following NSA (Yuan et al., 2025), we set the compression block size $l_{C_1} = 32$, stride $s_{C_1} = 16$, and attention block size $B = 64$. For our efficient block selection implementation in Section 3.4, we additionally set the LSE Approximation block size $l_{C_2} = 128$ and stride $s_{C_2} = 64$. We set the selected block count $|\mathcal{I}| = 96$ (including $|\mathcal{I}_{\text{init}}| = 1$, $|\mathcal{I}_{\text{topk}}| = 63$, and $|\mathcal{I}_{\text{local}}| = 32$) for both training and inference. Therefore, the total number of visible tokens is $|\mathcal{I}| \cdot B = 6$k. We conduct long-sequence finetuning on the pretrained model using 5B tokens, with an initial learning rate of 3e-4 and linear decay to 2.75e-4. The training batches contain sequences from four length intervals: 0-4k, 4-12k, 12-24k, and 24-32k, with token counts in a 1:1:1:1 ratio.

**Baselines.**   We finetune a baseline model with full attention, marked as FULLATTN, using the same training configuration as INFLLM-V2 (SPARSE). We then apply several typical training-free sparse attention methods on FULLATTN as baselines, including InfLLM (Xiao et al., 2024a) and MInference (Jiang et al., 2024). In addition, we present the results of SHORT with YaRN (Peng et al., 2023) to extend the context window size. In terms of trainable sparse attention, we compare with NSA (Yuan et al., 2025). By using the same training settings as in INFLLM-V2 (SPARSE), we finetune our pretrained model into an NSA version. We initialize NSA's three sets of KV parameters by replicating the original KV parameters in dense attention. As NSA does not publish their code, we adopt an open-source Triton implementation of NSA for experiments[2].

Table 1: Task Performance on RULER. Best results in sparse attention are bolded.

| Method | SG1 | SG2 | SG3 | MK1 | MK2 | MK3 | MV | MQ | VT | CWE | FWE | QA1 | QA2 | Avg. |
|---|---|---|---|---|---|---|---|---|---|---|---|---|---|---|
| FULLATTN | 100.00 | 100.00 | 100.00 | 96.00 | 94.00 | 92.00 | 82.00 | 98.50 | 93.20 | 44.40 | 91.33 | 48.00 | 56.00 | 84.26 |
| SHORT+YARN | 98.00 | 68.00 | 50.00 | 46.00 | 6.00 | 0.00 | 32.00 | 31.50 | 36.00 | 21.40 | 87.33 | 26.00 | 26.00 | 40.63 |
| INFLLM | 98.00 | 6.00 | 4.00 | 10.00 | 10.00 | 10.00 | 9.00 | 7.50 | 70.00 | 16.00 | 80.67 | 18.00 | 24.00 | 27.94 |
| MINFERENCE | 100.00 | **100.00** | **100.00** | 76.00 | 36.00 | 46.00 | 79.50 | 93.50 | 88.00 | **64.20** | **92.67** | 32.00 | **44.00** | 73.22 |
| NSA | **100.00** | 88.00 | 82.00 | 54.00 | 38.00 | 30.00 | 59.00 | 61.50 | 56.00 | 34.40 | 86.00 | 56.00 | 34.00 | 59.92 |
| INFLLM-V2 (SPARSE) | | | | | | | | | | | | | | |
|   w/  *LSE Approx* | **100.00** | **100.00** | **100.00** | **94.00** | **82.00** | 62.00 | **98.50** | 94.50 | **98.00** | 50.40 | 82.67 | **72.00** | 40.00 | **82.62** |
|   w/o *LSE Approx* | **100.00** | **100.00** | **100.00** | 92.00 | 80.00 | **64.00** | **98.50** | **95.50** | **98.00** | 47.80 | 81.33 | 70.00 | 40.00 | 82.09 |
| INFLLM-V2 (DENSE) | 100.00 | 100.00 | 100.00 | 94.00 | 98.00 | 98.00 | 99.00 | 98.00 | 98.40 | 52.80 | 90.00 | 76.00 | 44.00 | 88.32 |

---

[2]https://github.com/XunhaoLai/native-sparse-attention-triton

Table 2: Task Performance on LongBench and LongPPL. Best results in sparse attention are bolded.

| Benchmark | FULLATTN | SHORT+YARN | INFLLM | MINFERENCE | NSA | INFLLM-V2 (SPARSE) | INFLLM-V2 (DENSE) |
|---|---|---|---|---|---|---|---|
| LongBench ↑ | 42.30 | 37.86 | 32.30 | 41.55 | 37.10 | **42.54** | 42.49 |
| LongPPL ↓ | 2.06 | 5.28 | 12.01 | 2.62 | 4.24 | **2.12** | 2.00 |

For all the above sparse attention methods, we maintain the same sparsity level to ensure a fair comparison. We provide the training curve for trainable methods in Figure 5. NSA causes a disruption in the loss, while INFLLM-V2 is closer to FULLATTN.

## 4.2 TASK PERFORMANCE

In this section, we evaluate InfLLM-V2 and other baselines across various tasks. Notably, while the original NSA paper demonstrates performance comparable to full attention when training on long sequences from scratch, NSA fails to achieve satisfactory results in short-to-long adaptation settings. *This indicates that the substantial parameter overhead introduced by* NSA *renders it unsuitable for the conventional "pretraining-on-short, finetuning-on-long" paradigm.*

**Long-Context Understanding.** To evaluate InfLLM-V2's performance on long-input tasks, we compare InfLLM-V2 and different baselines on RULER (Hsieh et al., 2024), LongBench (Bai et al., 2024) and LongPPL (Fang et al., 2025). RULER is a synthetic dataset with a configurable average length. LongBench is a bilingual benchmark for long-context understanding. Compared to RULER, LongBench is primarily built from existing, real-world datasets. LongPPL is a perplexity evaluation benchmark for long sequences. The experimental results of RULER when the length is 32k are shown in Table 1. The results on LongBench and LongPPL are shown in Table 2. Please refer to Appendix C for detailed performance of the sub-tasks in LongBench. From the results, we can observe that: 1) INFLLM-V2 achieves the best performance compared to other sparse methods, with its results closely matching the strong, FULLATTN baseline. Alternative approaches, whether applying training-free sparsity or training-based sparsity, result in a substantial drop in performance. 2) Compared to NSA, INFLLM-V2 can achieve performance improvements through minimal finetuning on long-sequences. Although NSA has low training loss, its high perplexity on the LongPPL evaluations indicates that NSA has not adequately learned long-range dependencies. 3) A unique advantage of INFLLM-V2 is the flexibility to seamlessly switch between dense mode and sparse mode. This not only provides an option for dense computation but can also lead to further performance improvement, surpassing even the full attention baseline. 4) Furthermore, the INFLLM-V2 (SPARSE) variant with LSE Approximation does not lose any performance. Given its comparable performance and efficiency, we adopt the LSE Approximation setting for all subsequent tasks by default.

**Long Reasoning.** To evaluate the performance of InfLLM-V2 in long-output scenarios, we compared several major Long Reasoning tasks, including

Table 3: Task Performance on Long Reasoning Tasks.

| Method | MATH-500 | AIME 24 | AIME 25 | LCB v5 | LCB v6 | Avg. ↑ |
|---|---|---|---|---|---|---|
| FULLATTN | 86.00 | 37.50 | 30.63 | 30.67 | 29.14 | 42.79 |
| NSA | 83.80 | 28.75 | 23.54 | 25.15 | 25.14 | 37.28 |
| InfLLM-V2 (Sparse) | 87.80 | 38.33 | 29.38 | 29.94 | 27.83 | 42.66 |
| InfLLM-V2 (Dense) | 86.40 | 36.67 | 23.33 | 29.94 | 26.29 | 40.53 |

MATH-500 (Hendrycks et al., 2021b), AIME (MAA), and LiveCodeBench (LCB) (Jain et al., 2025). We finetune InfLLM-V2 and baselines on OpenMathReasoning (Moshkov et al., 2025) and OpenCodeReasoning (Ahmad et al., 2025). As InfLLM and MInference primarily accelerate long-input processing, we exclude them from this long-output evaluation. The experimental results are shown in Table 3. The results show that InfLLM-V2 attains performance on par with full attention, confirming its effectiveness for long-output scenarios.

**General Tasks.** We verify that the InfLLM-V2 architecture can freely switch back to Dense mode without performance degradation on short-sequence tasks after long-sequence fine-tuning. Zero-shot evaluations on MMLU (Hendrycks et al., 2021a), MMLU-Redux (Gema et al., 2025), CE-val (Huang et al., 2023), MATH-500 (Hendrycks et al., 2021b), HumanEval (Chen et al., 2021), MBPP (Austin et al., 2021) and BBH (Suzgun et al., 2023) are shown in Table 4. Experimental results show that InfLLM-V2 achieves performance comparable to full attention.

## 4.3 EFFICIENCY

We first evaluate the efficiency of our kernel implementation on NVIDIA A100 and NVIDIA 4090. We evaluate InfLLM-V2's inference efficiency on the batch=1 setting. We select FlashAttention-

Table 4: Task Performance on General Tasks.

| Method | MMLU | MMLU-Redux | CEval | MATH-500 | HumanEval | MBPP | BBH | Avg. ↑ |
|---|---|---|---|---|---|---|---|---|
| SHORT | 72.73 | 72.71 | 76.17 | 54.40 | 70.73 | 75.49 | 51.90 | 67.73 |
| FULLATTN | 73.38 | 70.24 | 78.11 | 54.60 | 71.34 | 75.10 | 49.13 | 67.41 |
| NSA | 68.27 | 66.39 | 74.33 | 44.40 | 62.20 | 65.00 | 43.81 | 60.63 |
| InfLLM-V2 (Dense) | 71.29 | 69.73 | 77.70 | 54.80 | 73.17 | 73.54 | 47.09 | 66.76 |

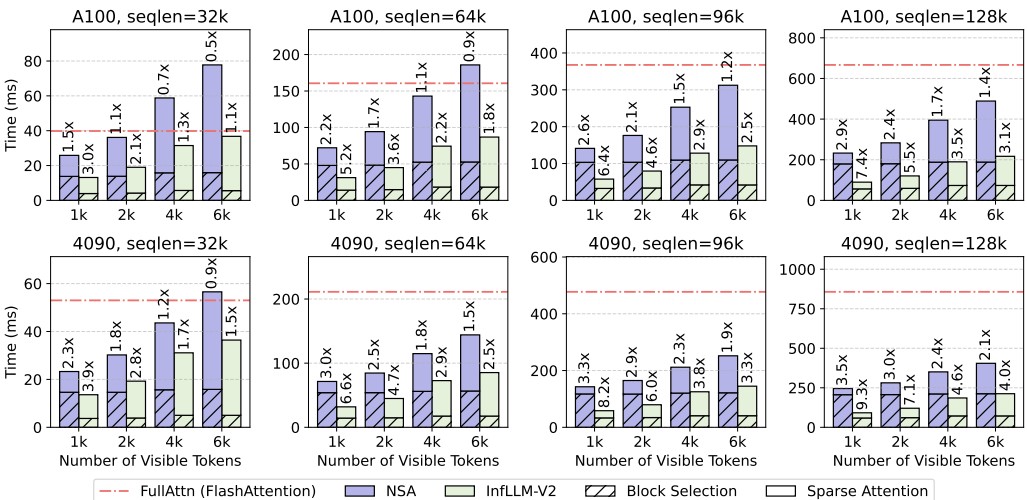

Figure 6: Speed of the kernels on NVIDIA A100 and NVIDIA 4090. "Number of Visible Tokens" means the number of key/value tokens each query token can attend to.

2 (Dao, 2024) implementation for full attention. For a fair efficiency comparison with NSA, we ignore its sliding attention component, and compare solely on the compression and sparse attention parts by selecting an equal number of blocks $|\mathcal{I}|$. Experiment results are shown in Figure 6. When the number of selected blocks is 16, InfLLM-V2 achieves up to $7.4\times$ over FlashAttention on A100 and $9.3\times$ on 4090. In contrast, NSA's speedup is limited to $3.5\times$ in the same setting. The breakdown of the execution time shows that the overhead from the *Block Selection* stage is greatly optimized by our efficient implementation in Section 3.4. We further conduct an ablation study on the *Block Selection*, as shown in Table 5, which shows the effectiveness of our proposed *LSE Approximation*.

Table 5: Ablation study of *Block Selection* efficiency, with and without *LSE Approximation*. All measurements are in time (ms), and the number of selected blocks is set to 16.

| Device | A100 | | | | 4090 | | | |
|---|---|---|---|---|---|---|---|---|
| Sequence Length | 32k | 64k | 96k | 128k | 32k | 64k | 96k | 128k |
| w/o LSE Approximation | 4.67 | 18.20 | 42.46 | 75.36 | 4.89 | 19.95 | 46.51 | 83.26 |
| w/ LSE Approximation | **3.93** | **14.07** | **32.44** | **56.59** | **3.70** | **14.39** | **33.16** | **59.04** |

The end-to-end inference speed (with a $|\mathcal{I}| = 96$ and W4A16 quantization (Frantar et al., 2025)) is shown in Figure 7. InfLLM-V2 can achieve $2.13\times$ prefilling speedup and $2.32\times$ decoding speedup. Since InfLLM-V2 does not accelerate the Feed-Forward Network (FFN) layers, a higher speedup ratio can be achieved by incorporating FFN-specific acceleration techniques in future work.

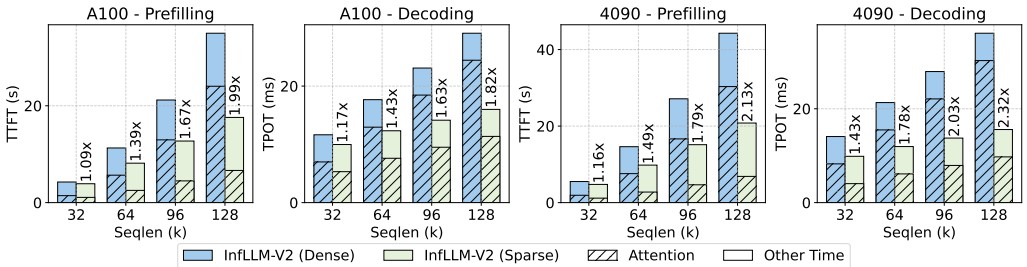

Figure 7: End-to-end inference speed of our 8B model when the number of visible tokens is 6k. TTFT means time-to-first-token, and TPOT means time-per-output-token.

## 5 CONCLUSION

In this paper, we introduced InfLLM-V2, a dense-sparse switchable attention framework designed to overcome the limitations of existing trainable sparse attention mechanisms. By ensuring architectural alignment with the standard pretrain-on-short and finetune-on-long workflow, InfLLM-V2 facilitates a seamless and efficient sparse adaptation to long contexts without requiring extra parameters or causing disruptive distributional shifts. We believe InfLLM-V2 offers a practical and powerful solution for advancing the capabilities of large language models in the long-context era.

## ACKNOWLEDGMENTS

This work is supported by the National Key Research and Development Program of China (2024YFB4505603), National Natural Science Foundation of China (No. 62576186) and the high-quality development project of MIIT.

## ETHICS STATEMENT

Our method has no potential risk since we focus on developing a novel algorithm.

## REPRODUCIBILITY STATEMENT

We provided our kernel implementation in the supplementary materials.

## THE USE OF LARGE LANGUAGE MODELS

After writing the paper manually, the paper was polished for clarity and readability using a large language model (LLM).

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

## A   THE USE OF LARGE LANGUAGE MODELS

After writing the paper manually, the paper was polished for clarity and readability using a large language model (LLM).

## B   IMPLEMENTATION DETAIL

We have shown the implementation of *Block Selection* in Section 3.4. We show the implementation detail of *Sparse Attention* here in Algorithm 2.

---

**Algorithm 2** Computation of *Sparse Attention*. (Suppose $h_{kv} = 1$ without loss of generality.)

---

**Require:** $\mathbf{Q} \in \mathbb{R}^{n \times G \times d_h}, \mathbf{K}, \mathbf{V} \in \mathbb{R}^{n \times d_h}$. Block sizes $B_k$.
  Divide $\mathbf{Q}$ into $n$ blocks $\mathbf{Q}_1, \dots, \mathbf{Q}_n$ of size $G \times d_h$ each.
  Divide $\mathbf{K}, \mathbf{V}$ into $T_k = \lceil n/B_k \rceil$ blocks $\mathbf{K}_1, \dots, \mathbf{K}_{T_k}$ and $\mathbf{V}_1, \dots, \mathbf{V}_{T_k}$ of size $B_k \times d_h$ each.
  Divide $\mathbf{O} \in \mathbb{R}^{n \times G \times d_h}$ into $n$ blocks of size $G \times d_h$ each.
  Divide the log-sum-exp $lse$ into $n$ blocks of size $G$ each.
  **for** $i = 1, \dots, n$ (parallel) **do**
    Load $\mathbf{Q}_i$ from HBM to on-chip SRAM.
    On chip, initialize $\mathbf{O}_i^{(0)} = (\mathbf{0})_{G \times d_h}, \ell_i^{(0)} = (\mathbf{0})_G, m_i^{(0)} = (-\infty)_G$.
    **for** $j = 1, \dots, T_k$ (sequential) **do**
      **if** $\mathbf{K}_j$ in visible tokens (determined by the $|\mathcal{I}(i)|$ in Eq. 3) **then**
        Load $\mathbf{K}_j, \mathbf{V}_j$ from HBM to on-chip SRAM.
        On chip, compute attention scores $\mathbf{S}_{ij} = \mathbf{Q}_i \mathbf{K}_j^\top \in \mathbb{R}^{G \times B_k}$.
        On chip, compute $m_i^{(j)} = \max(m_i^{(j-1)}, \text{rowmax}(\mathbf{S}_{ij})) \in \mathbb{R}^G$.
        On chip, compute $\tilde{\mathbf{P}}_{ij} = \exp(\mathbf{S}_{ij} - m_i^{(j)}) \in \mathbb{R}^{G \times B_k}$.
        On chip, compute $\ell_i^{(j)} = \exp(m_i^{(j-1)} - m_i^{(j)})\ell_i^{(j-1)} + \text{rowsum}(\tilde{\mathbf{P}}_{ij}) \in \mathbb{R}^G$.
        On chip, compute $\mathbf{O}_i^{(j)} = \text{diag}(\exp(m_i^{(j-1)} - m_i^{(j)}))^{-1} \mathbf{O}_i^{(j-1)} + \tilde{\mathbf{P}}_{ij} \mathbf{V}_j$.
    On chip, compute $\mathbf{O}_i = \text{diag}(\ell_i^{(T_k)})^{-1} \mathbf{O}_i^{(T_k)}$.
    On chip, compute $lse_i = m_i^{(T_k)} + \log(\ell_i^{(T_k)})$.
    Write $\mathbf{O}_i$ to HBM as the i-th block of $\mathbf{O}$.
    Write $lse_i$ to HBM as the i-th block of $lse$.
  **return** the output $\mathbf{O}$ and the log-sum-exp $lse$.

---

**Algorithm 3** Computation of *Dense Attention*. (Suppose $h_{kv} = 1$ without loss of generality.)

---

**Require:** $\mathbf{Q} \in \mathbb{R}^{n \times G \times d_h}, \mathbf{K}, \mathbf{V} \in \mathbb{R}^{n \times d_h}$. Block sizes $B_q, B_k$.
  Divide $\mathbf{Q}$ into $T_q = G \times \lceil n/B_q \rceil$ blocks $\mathbf{Q}_1, \dots, \mathbf{Q}_{T_q}$ of size $B_q \times d_h$ each.
  Divide $\mathbf{K}, \mathbf{V}$ into $T_k = \lceil n/B_k \rceil$ blocks $\mathbf{K}_1, \dots, \mathbf{K}_{T_k}$ and $\mathbf{V}_1, \dots, \mathbf{V}_{T_k}$ of size $B_k \times d_h$ each.
  Divide $\mathbf{O} \in \mathbb{R}^{n \times G \times d_h}$ into $T_q$ blocks of size $B_q \times d_h$ each.
  Divide the log-sum-exp $lse$ into $T_q$ blocks of size $B_q$ each.
  **for** $i = 1, \dots, T_q$ (parallel) **do**
    Load $\mathbf{Q}_i$ from HBM to on-chip SRAM.
    On chip, initialize $\mathbf{O}_i^{(0)} = (\mathbf{0})_{B_q \times d_h}, \ell_i^{(0)} = (\mathbf{0})_{B_q}, m_i^{(0)} = (-\infty)_{B_q}$.
    **for** $j = 1, \dots, T_k$ (sequential) **do**
      Load $\mathbf{K}_j, \mathbf{V}_j$ from HBM to on-chip SRAM.
      On chip, compute attention scores $\mathbf{S}_{ij} = \mathbf{Q}_i \mathbf{K}_j^\top \in \mathbb{R}^{B_q \times B_k}$.
      On chip, compute $m_i^{(j)} = \max(m_i^{(j-1)}, \text{rowmax}(\mathbf{S}_{ij})) \in \mathbb{R}^{B_q}$.
      On chip, compute $\tilde{\mathbf{P}}_{ij} = \exp(\mathbf{S}_{ij} - m_i^{(j)}) \in \mathbb{R}^{B_q \times B_k}$.
      On chip, compute $\ell_i^{(j)} = \exp(m_i^{(j-1)} - m_i^{(j)})\ell_i^{(j-1)} + \text{rowsum}(\tilde{\mathbf{P}}_{ij}) \in \mathbb{R}^{B_q}$.
      On chip, compute $\mathbf{O}_i^{(j)} = \text{diag}(\exp(m_i^{(j-1)} - m_i^{(j)}))^{-1} \mathbf{O}_i^{(j-1)} + \tilde{\mathbf{P}}_{ij} \mathbf{V}_j$.
    On chip, compute $\mathbf{O}_i = \text{diag}(\ell_i^{(T_k)})^{-1} \mathbf{O}_i^{(T_k)}$.
    On chip, compute $lse_i = m_i^{(T_k)} + \log(\ell_i^{(T_k)})$.
    Write $\mathbf{O}_i$ to HBM as the i-th block of $\mathbf{O}$.
    Write $lse_i$ to HBM as the i-th block of $lse$.
  **return** the output $\mathbf{O}$ and the log-sum-exp $lse$.

---

Similar to FlashAttention (Dao, 2024), the algorithm divides the input into blocks. The differences are: 1) The FlashAttention block size $B_k$ of $\mathbf{K}$, should divide the sparse attention block size $B$. That is, $B$ should be a multiple of $B_k$. 2) The FlashAttention block of $\mathbf{Q}$ typically contains a single attention head and multiple tokens. We follow NSA (Yuan et al., 2025) to make it contain a group of attention heads of a single token, so that they can share the same sparse pattern. 3) The inner loop of the FlashAttention iterates over all blocks of $\mathbf{K}$, whereas our method's loop only covers the visible blocks of the sparse attention. We also show the FlashAttention implementation of Dense Attention to Algorithm 3 for reference.

## C BENCHMARK DETAILS

We provide the detailed results of the LongBench benchmark, mentioned in Table 2, in Table 6. Following LongBench (Bai et al., 2024), the "Overall" score is computed by the macro-average over the six task categories.

Table 6: Task Performance on LongBench. Best results in sparse attention are bolded.

| Category | | FULLATTN | SHORT + YARN | INFLLM | MINFERENCE | NSA | INFLLM-V2 (SPARSE) | INFLLM-V2 (DENSE) |
|---|---|---|---|---|---|---|---|---|
| Single-Doc QA | NarQA | 21.38 | 18.17 | **21.02** | 20.16 | 18.34 | 20.75 | 21.03 |
| | Qasper | 43.80 | 30.98 | 34.92 | 44.51 | 39.96 | **45.29** | 45.29 |
| | MFQA-en | 55.07 | 43.81 | 49.39 | **54.83** | 51.35 | 53.53 | 53.54 |
| | MFQA-zh | 57.26 | 54.51 | 51.75 | 57.00 | 59.06 | **59.33** | 59.64 |
| Multi-Doc QA | HotpotQA | 50.13 | 48.49 | 44.03 | 48.00 | 46.78 | **54.11** | 54.07 |
| | 2WikiQA | 39.54 | 32.71 | 30.58 | 36.22 | 35.33 | **37.86** | 37.86 |
| | MuSiQue | 24.68 | 23.22 | 17.85 | **22.87** | 16.97 | 21.74 | 21.24 |
| | Dureader | 33.54 | 33.00 | 33.01 | **33.94** | 33.62 | 33.39 | 33.29 |
| Summary | GovReport | 32.17 | 31.93 | 21.40 | **32.21** | 28.72 | 30.33 | 30.38 |
| | QMSum | 24.35 | 22.45 | 20.96 | **25.05** | 23.81 | 24.58 | 24.35 |
| | MultiNews | 26.70 | 26.46 | 22.90 | **26.50** | 25.02 | 25.71 | 25.75 |
| | VCSUM | 16.37 | 16.55 | 17.81 | 16.17 | **19.12** | 16.17 | 16.20 |
| Few-shot Learning | TREC | 45.00 | 65.50 | **61.00** | 43.50 | 23.50 | 22.50 | 24.00 |
| | TriviaQA | 84.35 | 85.67 | 75.78 | 81.93 | 83.95 | **84.22** | 84.22 |
| | SAMSum | 40.26 | 42.92 | 37.46 | 39.81 | 38.47 | **40.69** | 40.51 |
| | LSHT | 37.75 | 38.00 | 24.57 | **35.75** | 25.50 | 22.01 | 21.47 |
| Synthetic Task | PsgCount | 4.00 | 4.06 | 3.00 | 3.50 | 3.50 | **5.00** | 4.50 |
| | PsgRe-en | 86.50 | 20.75 | 19.00 | 85.00 | 66.00 | **92.00** | 91.00 |
| | PsgRe-zh | 90.50 | 42.00 | 43.00 | 90.50 | 68.00 | **90.50** | 90.50 |
| Code | LCC | 35.72 | 58.65 | 31.35 | 35.91 | 33.83 | **44.73** | 44.73 |
| | RepoBen-P | 35.00 | 43.93 | 30.72 | 34.17 | 34.95 | **44.62** | 44.76 |
| Overall ↑ | | 42.30 | 37.86 | 32.30 | 41.55 | 37.10 | **42.54** | 42.49 |

## D HYPERPARAMETERS ANALYSIS

In this section, we present analyses and discussions regarding the choice of key hyperparameters in our framework, specifically focusing on the number of selected blocks ($|\mathcal{I}_{\text{topk}}|$), local blocks ($|\mathcal{I}_{\text{local}}|$), initial blocks ($|\mathcal{I}_{\text{init}}|$), and the block size ($B$). These parameters serve as the core variables controlling the trade-off between model performance and computational efficiency. As shown in Table 7, increasing $|\mathcal{I}_{\text{topk}}|$ from 31 to 63 yields significant performance gains, whereas further increasing it to 95 offers only marginal improvements. Similarly, while larger $|\mathcal{I}_{\text{local}}|$ improves results, the marginal gain decreases as computational overhead grows. Consequently, we select $|\mathcal{I}_{\text{topk}}| = 63$ and $|\mathcal{I}_{\text{local}}| = 32$ as the optimal configuration for balancing efficiency and effectiveness.

Table 7: Hyperparameter analysis on the number of selected blocks and local blocks.

| $\mathcal{I}_{\text{init}}$ | $\mathcal{I}_{\text{topk}}$ | $\mathcal{I}_{\text{local}}$ | RULER ↑ | LongPPL ↓ |
|---|---|---|---|---|
| 1 | 63 | 32 | 82.62 | 2.12 |
| 1 | 31 | 32 | 76.14 | 2.36 |
| 1 | 95 | 32 | 84.89 | 2.05 |
| 1 | 63 | 16 | 82.08 | 2.15 |
| 1 | 63 | 64 | 83.45 | 2.09 |

Regarding other parameters, we set $|\mathcal{I}_{\text{init}}|$ to a minimal value following the established practice in StreamingLLM (Xiao et al., 2024b), as a small number of attention sinks is sufficient for stability. For the block size $B$, we adopt $B = 64$ following NSA (Yuan et al., 2025). This choice is motivated by hardware efficiency, as FlashAttention kernels typically require block sizes of 64 or 128 to maximize Tensor Core utilization and memory bandwidth.

