# OpenReview forum: "InfLLM-V2: Dense-Sparse Switchable Attention for Seamless Short-to-Long Adaptation"
_ICLR.cc/2026/Conference — ICLR 2026 Poster_

### Official Review · Reviewer_S847 · 2025-10-30

**Soundness:** 3
**Presentation:** 3
**Contribution:** 2
**Rating:** 4
**Confidence:** 4

**Summary:**

This paper introduced Dense-Sparse Switchable Attention, a sparse attention that adapts model from short to long sequences. DSSA starts from natively trainable sparse attention (SSA), which further modifications, including: 1. Shared Key-Value Projection. Instead of using three sets of KV, DSSA use a single shared set. 2. Remove compressed attention and only used it to get sparse selected attention and sparse sliding attention. Then replace the MLP with pooling function. They did experiments on several benchmarks and demonstrated the efficiency while maintaining the performance.

**Strengths:**

1. The proposed architecture is simple to follow. Just a few modification to existing NSA.
2. The performance is demonstrated across different benchmarks.
3. The training process is quiet efficient. Figure 6 presented a clear comparision between NDA and DSSA.

**Weaknesses:**

1. Since DSSA is based on NSA, then the most important baseline should be NSA. Even NSA is 27b total parameters and 3b active parameters, which is larger than DSSA implementation in this paper, it is important to evaluate that on similar benchmarks. For examples, the main table, table 1 in NSA paper presented many benchmarks, which are also major benchmarks for LLM. But DSSA did not evaluate on those.

2. In NSA paper, NSA achieved better performance than Full Attention as in table 2 of the paper. However, in your paper the NSA is much worse than LongBench, which leads to concerns about implementation of NSA.

3. Still, in Figure 6 of NSA paper, it is around 9x faster than full attention but in your case it is only 0.9x to 2.2x. The gap is too large to be ignored.

**Questions:**

1. In figure 1. portable is mentioned as an advantage. What is the meaning here other than reusable and aligned?
2. Since the author examed the speed on different gpu, then what are the software configureations like cuda version and pytorch version? Different version could lead to performance gap for other baselines.
3. What does number of visible tokens mean? Does it mean how many keys/values each query token can attend to?

---

> ### Author Response · Authors · 2025-11-21
>
> We sincerely thank the reviewer for the constructive feedback and for acknowledging the simplicity and effectiveness of our proposed DSSA framework. We address your concerns and questions point-by-point below.
>
> ## W1: Comparison with NSA on major benchmarks.
>
> Actually, we **have evaluated** our method on the major benchmarks you mentioned, as shown in **Table 4** of our paper. We have reported results on MMLU, MMLU-Redux, CEval, MATH-500, HumanEval, MBPP, and BBH, which significantly overlap with the benchmarks in Table 1 of the NSA paper.
>
> To further address your concern, we run experiments to include the remaining metrics from the NSA Table 1 that are not initially displayed. Then we list all the results as follows.
>
> | Model (Setting) | MMLU | MMLU-PRO | CMMLU | BBH | GSM8K | MATH | DROP | MBPP | HumanEval | Avg. |
> | :------- | :-----: | :-----: | :-----: | :-----: | :-----: | :-----: | :-----: | :-----: | :-----: | :-----: |
> | NSA (27B, Few-shot*) | 56.5 | 28.6 | 58.7 | **52.1** | 52.0 | 26.4 | 54.5 | 46.6 | 34.8 | 45.6 |
> | DSSA (8B, 0-shot) | **71.3** | **55.5** | **79.7** | 47.1 | **87.7** | **54.1** | **65.5** | **73.5** | **73.2** | **67.5** |
>
> \* : Results are cited from the NSA paper, which uses few-shot settings for most benchmarks.
>
> ## W2: NSA performance gap and implementation concerns.
>
> This observation actually highlights the core motivation of our paper. The impressive results reported in the NSA paper are achieved because they train the model using the NSA architecture **from scratch** (pre-training phase). In contrast, our work focuses on a more common and economical paradigm in both industry and academia: **short-to-long adaptation**. This involves taking a **mature** model that has already been pretrained with standard Dense Attention on short sequences and adapting it to long sequences. The poor performance of NSA in our experiments is not due to implementation errors, but rather reveals a fundamental limitation of the NSA architecture in this specific "pretrain-on-short, finetune-on-long" paradigm. NSA introduces architectural mismatches that disrupt the knowledge of a pre-trained dense model.
>
> On the other hand, we build our DSSA implementation by modifying the code logic of the NSA baseline, and we have rigorously verified the correctness of the code against the NSA paper's formulation. The performance drop is an architectural consequence of the adaptation setting, which we analyze extensively in the paper.
>
> ## W3: The gap in speedup ratios
>
> The original NSA paper does not release its official code. To ensure a fair comparison, we adopt a reputable open-source Triton implementation of NSA, and **then begin our modification based on it**.
>
> While the NSA paper details the sparse attention kernel, it omits critical implementation details for the block selection kernel. Without specific kernel optimization strategies, this triton NSA implementation suffers from frequent Read/Write operations to the GPU HBM, resulting in a significant I/O bottleneck. As illustrated by the **dashed regions in Figure 6**, the unoptimized block selection accounts for a substantial portion of the total runtime in the NSA baseline. Addressing this bottleneck is exactly one of our technical contributions. In **Section 3.4 and Algorithm 1**, we introduce specific optimizations, such as **Fused Head Group Summation** and **LSE Approximation**, to minimize HBM I/O. These techniques significantly reduce the computational overhead of block selection, boosting the speedup from 2.2× to 5.2×.
>
> We believe that bridging this gap by providing an open-source and hardware-efficient implementation is a valuable contribution to the community. We have identified further opportunities for kernel fusion in the last paragraph of Section 3.4 and plan to address them in future work.
>
> ## Q1: Meaning of "Portable".
>
> "Portable" refers to the **ease of transferability**. It is the result of the model being both "reusable" and "aligned". Specifically, it means that our architecture introduces minimal distributional shifts during the short-to-long adaptation process, resulting in a smoother loss curve. Furthermore, it implies that the model retains **the flexibility to switch back to Dense Attention mode** at any time without requiring retraining, ensuring efficiency across different deployment scenarios.
>
> ## Q2: Software configurations.
>
> We conduct our experiments using PyTorch 2.6.0 and CUDA 12.1.
>
> ## Q3: Meaning of "number of visible tokens".
>
> Your understanding is completely correct. This value directly determines the sparsity level and the computational cost. We have updated the figure caption in the revised paper to provide this explicit definition and avoid confusion.

---

### Official Review · Reviewer_owJK · 2025-10-30

**Soundness:** 4
**Presentation:** 3
**Contribution:** 3
**Rating:** 6
**Confidence:** 2

**Summary:**

This paper introduces Dense-Sparse Switchable Attention (DSSA), a novel trainable sparse attention framework designed to efficiently adapt Large Language Models (LLMs) from short-sequence pretraining to long-sequence finetuning. The authors identify critical limitations in prior trainable sparse attention methods like Natively Trainable Sparse Attention (NSA), namely the introduction of excessive parameters, architectural mismatches that destabilize training, and inefficiency on short sequences.

DSSA addresses these issues with three main contributions:

- Seamless Short-to-Long Adaptation: DSSA reuses the key-value (KV) projection parameters from the original dense attention model, eliminating the need for extra parameters. It fuses multiple sparse patterns into a unified sparse attention module, maintaining architectural consistency with standard attention. This leads to stable training and preserves knowledge from pretraining.
- Efficiency on All Sequence Lengths: The parameter-free design allows a model equipped with DSSA to seamlessly switch back to standard dense attention for short sequences without any overhead, while effectively using sparse attention for long sequences.
Hardware-Aware Efficient Implementation: The authors identify the block selection mechanism as a key bottleneck in sparse attention. They propose a novel, two-pass, FlashAttention-style CUDA kernel that significantly reduces I/O overhead by fusing computations in SRAM. An "LSE Approximation" technique further optimizes this kernel, reducing the computational cost of the two-pass approach.
- Experiments on long-context understanding and reasoning tasks show that DSSA achieves performance nearly on par with full attention (retaining over 98%) while being up to 4x faster on the attention component and providing significant end-to-end speedups.

**Strengths:**

1. Clear Problem Formulation: The paper does an excellent job of positioning its work. It clearly articulates the shortcomings of the existing state-of-the-art trainable sparse attention method (NSA) and directly proposes solutions for each identified problem (parameter overhead, training instability, and short-sequence inefficiency). The framing around the "pretrain-on-short, finetune-on-long" paradigm is highly relevant and practical.

2. Elegant and Simple Design: The core idea of reusing existing parameters and unifying sparse patterns is both simple and powerful. This design choice directly leads to several benefits: zero parameter overhead, stable finetuning (as evidenced by the loss curves in Figure 5), and the flexibility to switch between dense and sparse modes. This makes DSSA a much more practical and "drop-in" solution compared to more architecturally disruptive methods.

3. Strong Technical Contribution in Implementation: The work in Section 3.4 is a significant strength. Identifying the I/O bottleneck in the block selection step and engineering a custom CUDA kernel to solve it shows a deep understanding of hardware-aware optimization. The LSE Approximation is a clever trick to mitigate the cost of their two-pass solution. This focus on practical, low-level optimization is crucial for realizing the theoretical benefits of sparse attention and is a valuable contribution.

4. Comprehensive and Convincing Experiments: The empirical evaluation is thorough. The authors compare DSSA not only against full attention and NSA but also against popular training-free methods (InfLLM, MInference) and a context-extension method (YaRN).

**Weaknesses:**

While the paper is strong, there are a few areas that could be improved or clarified:
1.  Direct Comparison to NSA: The paper's primary baseline for trainable sparse attention is NSA. However, the authors note they used an open-source Triton implementation of NSA, as the official code was not available. This raises a potential concern about whether the performance drop observed for NSA is due to fundamental architectural flaws (as argued) or sub-optimal implementation/hyperparameters in the third-party version. The claims against NSA would be stronger if benchmarked against an official implementation or with more extensive hyperparameter tuning for the version used.
2.  Limited Exploration of Hyperparameters: The effectiveness of DSSA's sparse pattern depends on several hyperparameters, such as the number of initial, local, and top-k blocks (`N_init`, `N_local`, `|I_topk|`) and the block size (`B`). The paper fixes these values for the experiments but does not provide an ablation study on their impact. Understanding the sensitivity of the model's performance and efficiency to these choices would be valuable for practitioners.
3.  Generalizability to Other Architectures: The experiments are conducted on an 8B parameter model with a specific Grouped-Query Attention (GQA) configuration. While the principles of DSSA seem general, its effectiveness and the efficiency of its custom kernels might vary with different model sizes, head dimensions, or attention variants (e.g., Multi-Query Attention). Discussing the potential challenges or necessary adaptations for other architectures would enhance the paper's scope.
4.  Novelty of Component Ideas: The idea of merging local (sliding window) and global (top-k) attention patterns is not entirely new and exists in prior work like BigBird or Longformer. The paper's main novelty lies in how it frames this within a *trainable, zero-parameter* adaptation framework and the efficient implementation. The paper could be more precise in situating its contribution relative to these earlier sparse attention architectures.

**Questions:**

1.  Block Selection Robustness: The block selection mechanism relies on attention scores from a compressed, coarse-grained representation of the keys. Could you comment on the robustness of this selection? Are there scenarios or tasks where this coarse approximation might fail to identify crucial fine-grained information, leading to performance degradation?
2.  Training Stability Details: Figure 5 shows a compellingly smooth training curve for DSSA compared to NSA. Could you elaborate on why DSSA's fine-tuning is more stable? Is it solely due to the shared parameters, or does the unified single-output structure (as opposed to NSA's gated multi-output) also play a significant role?
3.  On the `DSSA-Dense` Performance: In several tables (e.g., Table 1 on RULER), the `DSSA-Dense` model (which is the finetuned model switched back to dense mode) outperforms the `FULLATTN` baseline that was also finetuned on long contexts. This is an interesting result. Do you have a hypothesis for why this "sparse-to-dense" training regimen leads to better performance than continuous dense training? Could the sparse fine-tuning be acting as a form of regularization?
4.  LSE Approximation Impact: The LSE Approximation is a key part of your efficiency gains, reducing the overhead of the two-pass approach from 2x to 1.25x. Table 1 shows that using the approximation does not hurt performance on RULER. Have you confirmed this holds true across all other tasks? Is there a theoretical or empirical bound on the error introduced by this approximation?

---

> ### Author Response · Authors · 2025-11-21
>
> We sincerely thank the reviewer for the insightful comments and for recognizing the practicality of our framing around the "pretrain-on-short, finetune-on-long" paradigm. Below, we address the specific concerns and questions raised.
>
> ## W1: Direct Comparison to NSA and implementation validity
>
> We primarily implement our DSSA based on the code logic of the NSA baseline. During this process, we rigorously verify our baseline against the NSA original paper to ensure correctness and logic alignment.
>
> We argue that NSA's performance drop under the "short-to-long adaptation" setting is **not due to implementation bugs**, but rather a fundamental **architectural mismatch**. NSA introduces substantial extra parameters, disrupting the pretrained feature distribution and causing the "loss fluctuations" seen in Figure 5. This instability makes NSA ineffective for short-to-long adaptation, regardless of the kernel implementation quality.
>
> ## W2: Analysis of hyperparameters
>
> Thank you for this valuable suggestion. We have added an analysis in **Appendix D** of our revised paper.
>
> 1. As shown in the new results, increasing $|I_{topk}|$ from 31 to 63 yields significant gains, but increasing it further to 95 provides marginal benefit while hurting efficiency. Thus, we selected $|I_{topk}|$=63 as the optimal balance between performance and efficiency. Similarly for $|I_{local}|$.
>
> 2. For $|I_{init}|$, we follow established practices (e.g., StreamingLLM), where a small fixed value is sufficient to anchor attention. For the block size $B$, we follow NSA to set $B$=64, since FlashAttention typically operates best with a block size of 64 or 128. $B$=64 provides high efficiency across a range of head dimensions.
>
> ## W3: Generalizability to other architectures
>
> 1. **For MQA:** Our block selection mechanism (Section 3.3) operates at the group level. A larger GQA group size means more query heads share the same selected KV blocks. This significantly improves I/O efficiency by reducing redundant memory reads and also increases computational parallelism.
>
> 2.  **For larger hidden sizes or head dimensions:** Scaling up the model increases the arithmetic intensity. This helps overlap the slight memory overhead caused by the sparse block iteration in sparse attention kernels, making DSSA even more efficient on larger models.
>
> ## W4: Novelty and relation to prior work
>
> We thank the reviewer for pointing this out. As you summarized, the main emphasis of our paper is indeed on the trainable and zero-parameter framework. We discuss merging sparse patterns just to distinguish our approach from the current state-of-the-art trainable baseline, NSA. In our revised paper, we have updated the **Related Work** section to clarify our novelty.
>
> ## Q1: Robustness of block selection: can coarse representations capture fine-grained details?
>
> To ensure robustness, we employ a **3-stage compression strategy** (Section 3.3). Crucially, early stages are designed to preserve **local fine-grained information**. After that, max-pooling ensures that critical local information can drive the final block score, acting as a high-recall filter. Our results on retrieval-heavy benchmarks (RULER, LongBench) confirm that this mechanism successfully captures fine-grained details and matches full attention performance.
>
> ## Q2: What drives the training stability: shared parameters or the unified single-output structure?
>
> Both factors contribute to DSSA's stability. Removing either one factor would reintroduce architectural mismatches or extra parameters. Together, these factors create a simpler optimization landscape that inherits and preserves the pretrained model's capabilities.
>
> ## Q3: Why does DSSA-Dense outperform Full Attention? (The "Sparse-to-Dense" Regularization Hypothesis)
>
> We also noted this interesting result and verified our evaluation pipeline repeatedly to ensure its validity. Our hypothesis aligns with your intuition. During the sparse adaptation phase, the model is "forced" to utilize only the most critical context blocks. This constraint **prevents the model from attending to noisy contexts**. When we switch back to dense mode (DSSA-Dense), the model retains these "denoised" attention capabilities while regaining access to the full context, resulting in superior performance. We plan to investigate this "sparsity-induced regularization" phenomenon more deeply in future work.
>
> ## Q4: Does LSE Approximation work on all benchmarks (other than RULER)?
>
> We apologize for the lack of clarity. In fact, **all our subsequent experiments** (LongBench, LongPPL, and Long Reasoning tasks) **utilized the LSE Approximation by default**. We have revised the experiment section to clarify this.
>
> As for the theoretical intuition you ask, the approximation only affects the log-sum-exp scalar used for softmax normalization. A slightly coarser estimation of the normalization factor preserves the relative ranking of attention scores.

---

### Official Review · Reviewer_2Rsd · 2025-11-02

**Soundness:** 3
**Presentation:** 3
**Contribution:** 3
**Rating:** 6
**Confidence:** 3

**Summary:**

This paper presents Dense Sparse Switchable Attention, abbreviated DSSA, a trainable sparse attention framework designed for smooth short to long sequence adaptation in transformer models. The core idea is to reuse the pretrained dense key value parameters for sparse attention, to merge selected block and sliding window patterns into a unified sparse mask, and to replace learned compression with a parameter free multi stage pooling scheme. The paper also proposes engineering optimizations including a fused head group summation and a log sum exp approximation to reduce GPU memory traffic and compute during block selection. Experimental results on long context understanding and long chain of thought reasoning show that DSSA keeps model performance close to dense attention while delivering substantial speedups. Reported highlights include up to 4 times faster end to end inference relative to dense attention with about 98 percent to 99.7 percent task performance retention and kernel level speedups of up to about 7 to 9 times versus FlashAttention in some settings.

**Strengths:**

The paper addresses an important and timely problem for large language models namely efficient and stable adaptation from short pretraining to long finetuning. The proposed design is conceptually appealing because it removes the architectural mismatch between dense and sparse attention by reusing existing dense KV projections and producing a single sparse attention output rather than multiple gated outputs. This design choice improves compatibility with the standard pretrain on short then finetune on long workflow and appears to reduce training instability. The technical contributions combine modeling and systems innovations in a way that is practical. On the modeling side the union of selected and sliding patterns into a single mask and the three stage coarse to fine pooling are novel and justified by the need to preserve granularity while remaining block sparse. On the systems side the fusion of group summation into an SRAM friendly loop and the LSE approximation are concrete optimizations that address real GPU IO bottlenecks and yield large empirical speedups. The empirical evaluation is comprehensive in scope and includes long input comprehension, long reasoning and general short sequence tasks, which supports the claim that DSSA preserves accuracy while improving latency. The paper is generally clear and well organized with algorithmic pseudocode, explicit hyperparameter choices, and a sensible set of baselines including dense attention and several sparse methods.

**Weaknesses:**

1. The paper promises to "unlock the full potential of sparse attention" but does not discuss the limits of DSSA in terms of model scale and when the FFN or other layers dominate end to end costs.
2. The training stability claim is interesting but under analyzed. The loss disruption observed for NSA is shown but the paper should quantify downstream performance variance, report random seed statistics, and show whether DSSA reduces catastrophic forgetting when switching modes.
3. The NSA comparison can be fragile because NSA is reimplemented and initialized by parameter replication. Authors can provide more details that ensure a fair comparison such as exact initialization, learning rate schedules, and etc.

**Questions:**

See above.

---

> ### Author Response · Authors · 2025-11-21
>
> We thank the reviewer for the constructive feedback and for recognizing our method as "conceptually appealing" and our system optimizations as "practical" and "concrete". Below, we address the specific concerns raised.
>
> ## W1: Analysis on the limits of non-attention overhead.
>
> We agree with the reviewer that other components, such as FFN layers, are also important.
> However, for the long-context scenarios that DSSA targets, attention remains the dominant cost. As illustrated by the **hatched regions** in Figure 7, the **Attention mechanism constitutes the dominant portion** of the computational load. This confirms that attention is the apparent bottleneck in long-sequence scenarios, and optimizing it leads to a significant end-to-end speedup.
>
> We have also explicitly acknowledged the limit regarding non-attention layers in our original paper (Section 4.3): "a higher speedup ratio can be achieved by incorporating FFN-specific acceleration techniques in future work." Optimizations for FFNs (e.g., sparse activation or MoE) are orthogonal to our contribution and can be combined with DSSA in future research.
>
> ## W2: Clarification on "training stability" and analysis of catastrophic forgetting.
>
> We thank the reviewer for pointing out the ambiguity regarding the term "stability".
> Our intended meaning is that the architectural mismatch causes severe loss fluctuations and disrupts the smooth adaptation process from short to long contexts. To improve clarity, we have **revised** the introduction section in our revised paper. We have replaced terms like "training instability" with more precise descriptions, such as "hinders smooth adaptation" and "minimizing loss fluctuations".
>
> Regarding **catastrophic forgetting**, we have provided this analysis in Table 4 in our original paper. This table evaluates the performance on short-sequence benchmarks (MMLU, HumanEval, etc.) after the models have finished long-context finetuning. DSSA-Dense (switching back to dense attention after long-context finetuning) retains performance comparable to the original SHORT baseline, demonstrating minimal forgetting. In contrast, NSA shows significant degradation, indicating that its architectural mismatch leads to a loss of general capabilities during adaptation.
>
> ## W3: Fairness of the NSA baseline comparison.
>
> We want to clarify that the performance drop in NSA is not due to our implementation, but due to the architectural mismatch before and after adapting the pretrained model. We explain the fairness of our NSA baseline settings in three steps:
>
> 1. Training workflow: NSA is designed to be trained from scratch. In our paper, we follow the standard industry workflow to build long-context models: "pretrain-on-short, finetune-on-long."
>
> 2. Parameter initialization: NSA requires three sets of KV parameters (block selection attention, sparse attention, and sliding-window attention), but a standard dense model only has one. When switching from Dense to NSA, we have to initialize the additional parameters.
>
> 3. Why we copy parameters: We believe replicating the parameters is the fairest approach. The NSA paper[1] itself claims that dense attention naturally contains sparse patterns. Copying the parameters allows the model to reuse the sparse patterns it has already learned (for block selection attention, sparse attention, and sliding-window attention). If we use random initialization for the additional KV parameters, it would introduce excessive noise and even further disrupt training.
>
> Therefore, the loss fluctuations and performance drop we observed are not artifacts of unfair comparison. They simply demonstrate that NSA's complex multi-branch structure is difficult to adapt from a pretrained dense model, whereas DSSA is designed specifically for this transition.
>
> [1] Yuan, Jingyang, et al. "Native sparse attention: Hardware-aligned and natively trainable sparse attention."

---

### Comment · Area_Chair_PvnA · 2025-11-25

Dear Reviewers,
Thank you to those who have already begun interacting with the authors — your timely follow-ups are greatly appreciated and reflect the professionalism and care that uphold our community’s standards.
For reviewers who have not yet responded, I would like to offer a gentle reminder. Authors have invested substantial time and effort into preparing their rebuttals, carefully addressing each concern raised in the reviews. As fellow researchers, we all understand the importance of being heard and having our clarifications considered. Even a brief acknowledgment or follow-up question helps ensure that the evaluation remains fair, thorough, and respectful of everyone’s work.
Your engagement during this phase is essential for maintaining a constructive and high-quality review process. Thank you again for your service and for treating both your fellow reviewers and the authors with the same consideration you would hope to receive.
Best regards,
AC

---

### Meta-Review · Area_Chair_vMmX · 2026-01-03

**Summary:**

This paper introduces DSSA, a sparse attention to handle long sequence modeling, based on NSA architecture. All reviewers agree the contributions of this paper, that removes architectural mismatch between dense and sparse attention with simple operations, and easy to follow. In parallel, reviewers also raises these concerns after the reviewing stage:

1. Limitations on Model Scale and other Layers. (Reviewer **2Rsd**)
2. Training stability  (Reviewer **2Rsd**, **owJK**)
3. Mismatch on NSA (Reviewer **2Rsd**, **owJK**, **S847**)
4. Exploration of Hyperparameters (Reviewer **owJK**)
5. Generalizability (Reviewer **owJK**)
6. Component (Reviewer **owJK**)

In general, the most severe issues in this paper is about the mismatch of NSA implementation as NAS does not release official code. For all other questions, authors have provided respones to address them. Overall, this paper is simple and good to follow. Therefore, I suggest acceptance for this paper.

**Reviewer Concerns:**

Authors have addressed most of reviewers' concerns among limitations, stability and other explorations like hyper-parameters. One remaining concern is about NSA mismatch. However, considering that NSA does not release official code and authors have provided code in the supplemental material. I think most of concerns have been addressed.

**Reviewer Scores:**

The major issue for all reviewers is about the NSA implementation. After reviewing authors' feedback, I think reviewers could increase after checking the code.

---

### Decision · Program_Chairs · 2026-01-26

Accept (Poster)